# Antibacterial Properties of Coaxial Spinning Membrane of *Methyl ferulate/zein* and Its Preservation Effect on Sea Bass

**DOI:** 10.3390/foods10102385

**Published:** 2021-10-08

**Authors:** Tingting Li, Yue Shen, Haitao Chen, Yuchen Xu, Dangfeng Wang, Fangchao Cui, Yujuan Han, Jianrong Li

**Affiliations:** 1Key Laboratory of Biotechnology and Bioresources Use, Dalian Minzu University, Dalian 116000, China; jwltt@dlnu.edu.cn; 2College of Food Science and Technology, Bohai University, Jinzhou 121000, China; 2019015053@qymail.bhu.edu.cn (Y.S.); 2020015165@qymail.bhu.edu.cn (Y.X.); cuifangchao1@qymail.bhu.edu.cn (F.C.); 2019015007@qymail.bhu.edu.cn (Y.H.); 3Beijing Key Laboratory of Flavor Chemistry, Beijing Technology and Business University (BYBU), Beijing 100048, China; chenht@th.btbu.edu.cn; 4School of Food Science and Technology, Jiangnan University, Wuxi 214122, China; 7190112089@stu.jiangnan.edu.cn

**Keywords:** electrospinning, methyl ferulate, sea bass, preservation

## Abstract

Methyl ferulate is a new natural antibacterial agent with strong activity and low toxicity. It has good application prospects in food preservation. In this paper, the antibacterial activity of methyl ferulate against *Shigella putrefaciens* was verified, and it was embedded into zein by electrospinning technology to prepare fiber membranes. The addition of methyl ferulate could improve the tensile strength of zein fiber membrane and decrease the crystallinity of the membrane, which was mainly a physical combination. The fiber membrane improved the thermal stability of methyl ferulate. The water contact angle (WCA) decreased to 54.85°. The results showed that methyl ferulate in fiber membrane could be released slowly, gradually exerting its antibacterial activity. After coating perch with *methyl ferulate/zein* fiber membrane, the growth of microorganisms in perch meat was inhibited, and the pH value and total volatile basic nitrogen (TVB-N)content were effectively increased. In a word, methyl ferulate had antibacterial activity in the fiber film, which was able to achieve a sustained release effect in the process of fish packaging, prolonging its antibacterial activity, and having preservation effect on sea bass; thus, it could be used in food packaging.

## 1. Introduction

Sea bass has a high protein content, is rich in essential amino acids, has a delicious taste, and contains high levels of unsaturated fatty acids and trace elements [1]. It is an ideal food material for obtaining high-quality protein [2]. However, due to the growth of spoilage microorganisms and endogenous enzymes that promote the hydrolysis of muscle proteins and oxidation of fatty substances, fresh sea bass is prone to spoilage. Therefore, the short shelf life of fresh sea bass hinders the circulation and sale of products and reduces its commercial value [3]. Therefore, it is an urgent problem to improve the preservation quality and extend the shelf life of sea bass.

To improve the safety of aquatic products and prolong their shelf life, electrospinning technology is used to combine polymers with antioxidant and antibacterial agents to develop bioactive food packaging, which is conducive to inhibiting fish spoilage and oxidation [4,5,6]. Electrospinning technology can produce fiber membranes by applying a high-voltage electrostatic field to the jet of a polymer solution. When a critical voltage is applied, the charges gather on the surface of the liquid, and the coulomb repulsion of the charges overcomes the surface tension of the polymer droplets. The spinning solution is ejected into a Taylor cone shape, and the solvent volatilizes to form electrospun fibers [7]. This process ensures a nano/microstructure with a large area/volume ratio and a large number of internal pores, high packaging efficiency, and enhanced protection of the physical and functional properties of compounds [8]. Owing to the high specific surface area and high porosity of electrospun fibers, electrospinning has been used to produce edible nanofibers to encapsulate antibacterial substances to maintain the quality of fruits and meats [9].

Zein is a kind of water-insoluble hydrophobic plant protein extracted from corn, which has the advantages of good biocompatibility, biodegradability, excellent film-forming performance, high heat resistance, oxygen resistance, and is considered to be a potential functional component embedding material [10,11]. More and more researchers have used zein to prepare antibacterial nanofibers for food preservation. Yao et al. [12] encapsulated rose hip seed oil into zein nanofibers to improve the shelf life of bananas. Altan et al. [13] encapsulated carvacrol into zein nanofibers to improve its sustainable release properties, and the nanofibers were used as active packaging for bread preservation. Methyl ferulate is a derivative of ferulic acid, which can reduce oxidation activity. Owing to its low toxicity, it can be sufficiently used in skin care products, cosmetics, health care products, medical treatment, food additives, and other fields [14,15,16]. It has recently been widely used for the prevention of food oxidation.

In this study, a coaxial electrospinning process was developed to encapsulate methyl ferulate in zein and prepare a *methyl ferulate/zein* coaxial electrospinning membrane. One of the advantages of coaxial electrospinning is that it can encapsulate non-electrospun functional macromolecules, such as proteins and drugs, in a simple one-step process, encapsulate or protect these macromolecules from the influence of the external environment, control the release rate of these macromolecules from the core, and reduce the problem of initial burst release [17,18]. Ding et al. [19] prepared a coaxial poly (lactic acid propyl gallate) fiber film by PLA embedding propyl gallate. The release of propyl gallate wrapped in electrospun fibers was relatively slow, reaching 86.46% after 192 h. Coaxial electrospun fibers played an effective role in drug delivery, and the morphology, mechanical properties, crystal structure, molecular structure, thermal properties, WCA, and sustained-release properties of the membrane were studied. The total viable counts (TVC), pH value, TVB-N content, and lipid oxidation of sea bass were measured to study the effect of the fiber film on the preservation of sea bass. The purpose of this study was to develop a fresh-keeping packaging method that can effectively extend the cold storage period of sea bass and promote the application of electrospun fiber film to provide a reference for the preservation of other fish.

## 2. Materials and Methods

### 2.1. Materials

Acetic acid, ethanol, and methanol (analytical purity) from Tianjin Jindong Tianzheng Fine Chemical Reagent Factory; zein (≥95%) from Shanghai Yuanye Biotechnology Co., Ltd., Shanghai, China; methyl ferulate (≥95%) from Haohong Biomedical Technology Co., Ltd., Shanghai, China; and electrospinning equipment, ET-2535H, from Beijing Yongkang Leye Technology Development Co., Ltd., Beijing, China.

### 2.2. Characterization of the Electrospun Fibers

#### 2.2.1. Electrospinning Process

(1) Preparation of the spinning solution: A certain amount of zein was weighed and dissolved in a mixed solution of acetic acid and ethanol (volume ratio: 3:7) to prepare a 30% electrospinning solution. The spinning solution was stirred on a constant temperature magnetic stirrer at room temperature for 3–4 h until the zein was completely dissolved. The solution was allowed to stand for 1 h to remove bubbles and presented as a yellow transparent liquid with a certain viscosity. Methyl ferulate was dissolved in methanol to prepare a light-yellow transparent solution of 60 mg/mL.

(2) Single-axis zein electrospinning: 4 mL of zein spinning solution was transferred to a 5 mL syringe and placed on a syringe pump. The aluminum foil paper was wrapped on the receiving cylinder, the distance between the receiving cylinder and needle mouth was 15 cm, the receiving speed of the receiving cylinder was 10 r/min, the translational stroke of the syringe was 60 mm, the liquid flow rate was 0.75 mL/h, and the applied voltage to the spinning solution was 15 kV. The fiber film was peeled from the aluminum foil and dried overnight to remove the residual solvent.

(3) Coaxial electrospinning of *methyl ferulate/zein*: As shown in Figure 1, 4 mL of zein spinning solution was transferred to a 5 mL syringe and used as a shell solution with a flow rate of 1 mL; 1 mL of methyl ferulate was used as the core solution and the liquid flow rate was 0.25 mL/h. The distance between the receiving cylinder and needle mouth was 15 cm, the receiving speed of the receiving cylinder was 10 r/min, and the voltage of the spinning solution was 24 kV. The fiber film was peeled from the aluminum foil and dried overnight to remove the residual solvent. The theoretical concentration of methyl ferulate in the fiber membrane was 47.6 mg/g.

#### 2.2.2. Morphology Characterization

Preparation of scanning electron microscopy (SEM, SS-4800, Hitachi, Japan) samples: The obtained fiber film was cut into 5 mm × 5 mm. After gilding, the 5-mm thin sheet was observed by SEM, and the diameter of the electrospun fibers in the electron microscope was measured using Image J software. Fifty electrospun fibers were selected for each sample to analyze the average and standard deviation of the fiber diameter.

Preparation of transmission electron microscopy (TEM, Tecnai G2 F30, FEI company, Hillsboro, State of Oregon, USA) samples: A small amount of fiber membrane was obtained with tweezers and placed on a copper mesh; 10 μL of deionized water was drawn into the copper mesh with a pipette gun. The fiber membrane and copper mesh were fixed. The copper mesh was dried naturally for 1 h. Then, the fiber morphology was observed using a transmission electron microscope.

#### 2.2.3. Mechanical Properties Test

The fiber film was cut into a 70 mm × 25 mm sample. The tensile strength of the electrospun fibers was measured using a tensile probe on a texture analyzer (TA-XT Plus, Lotun Science Co.,Ltd., Beijing, China). The test parameters were as follows: the test length was 40 mm, the tensile rate was 5 mm/min, the tensile distance was 70 mm, and the test was conducted at room temperature. Based on these tests, the tensile strength (σt) and elongation at break of the (εt) electrospun fibers were calculated using Formulas (1) and (2) [19] as follows:(1)σt=p/(b×d)
where σ T is the tensile strength (MPa), *p* is the tensile force (N) at the fracture site, *b* is the width of the sample (mm), and *d* is the sample thickness (mm).
(2)$$\varepsilon t=\left(L-L_{0}\right)/L_{0}\times 100\%$$

Here, εt is the elongation at break (%), *L*_0_ is the original length of the sample (mm), and *L* is the length of the fracture (mm).

#### 2.2.4. Determination of X-ray Diffraction (XRD)

An appropriate amount of fiber film, methyl ferulate powder, and zein powder were used for XRD (Ultima IV, Rigaku, Japan). When the working voltage and current were 40 kV and 40 Ma, Cu Kα radiation was used. The scanning speed was set to 16 °/min and a 2θ range of 5–90° was used to determine the crystallinity of the electrospun fibers.

#### 2.2.5. Determination of Fourier Transform Infrared Spectroscopy (FT-IR)

The fiber membrane methyl ferulate powder and zeaxyl alcohol-soluble protein powder were dried under an incandescent lamp, cut into pieces, mixed with potassium bromide, and pressed into a tablet press to form a piece. The sample was analyzed by FT-IR (Alpha-Centauri 560, Thermo Fisher Nicolet, Waltham, MA, USA) in the wavenumber range of 400–4000 cm^−1^.

#### 2.2.6. Thermal Performance Analysis

Thermogravimetric analysis (TGA, PerkinElmer, Cleveland, OH, USA) was used to analyze the fiber membrane samples. After the fiber membrane was cut and dried, the appropriate 3–6 mg fiber membrane was weighed, placed in the sample tank, and then placed into the thermogravimetry for measurement. The temperature range of the analyzer was 40–800 °C, and the rate of temperature increase was 20 °C/min. The loss rate of the sample quality was recorded with an increase in temperature.

The thermal properties of the fiber films were analyzed using differential scanning calorimetry (DSC, Q2000, TA Instruments-Waters LLC, Shanghai, China). After drying the fiber membrane, 6–8 mg of the sample was placed in an aluminum crucible and protected by N_2_ at a temperature between 20 and 250 °C and a heating rate of 20 °C/min; the DSC curve of the sample temperature rise was then recorded.

#### 2.2.7. Determination of WCA

The WCA of the corn alcohol-soluble protein fiber membrane and the coaxial methyl ferulate/zeaxyl alcohol-soluble protein fiber membrane were measured using an optical contact angle meter (SL200KB, KINO, Boston, MA, USA). A 0.1-mm-thick fiber film was fixed to a 3-axis horizontal platform, and 10 μL of deionized water was fixed onto each membrane sample. When the water drops contacted the membrane for 2 s, the WCA of the sample was recorded.

#### 2.2.8. In Vitro Release Behavior

Methyl ferulate was dissolved in methanol and diluted twice with PBS to prepare methyl ferulate solutions of different concentrations. Then, the absorbance of methyl ferulate was determined using an ultraviolet spectrophotometer (UV2550, Shimadzu, Kyoto, Japan) at 254 nm. The concentration and absorbance of methyl ferulate were analyzed by linear regression analysis, and a fitting equation was obtained. 1 mL of PBS was added to a conical flask containing 10 mg of *methyl ferulate/zein* fiber membrane and transferred to a constant-speed oscillator (light avoidance, 4 °C, 30 rpm). Quantitative liquid samples were obtained regularly to determine the absorbance of the methyl ferulate. Meanwhile, the same amount of methyl ferulate was added to the flask. The concentration of methyl ferulate was calculated according to the standard curve, and the cumulative release was then calculated.

### 2.3. Coating Effects of the Electrospun Fibers on the Quality of Sea Bass during Cold Storage

#### 2.3.1. Verification of Antibacterial Activity

The antibacterial activity of *methyl ferulate/zein* fiber membrane was tested with *S**. putrefaciens* as the experimental strain. The culture was diluted to 10^6^ CFU/mL for analysis. Before use, both sides of the fiber were exposed to ultraviolet light for 10 min to eliminate surface contamination. Then, 100 mg of zein fiber and *methyl ferulate/zein* fiber were placed into a 2 mL culture medium inoculated with bacteria. The theoretical content of methyl ferulate in the fiber membrane was 4.76 mg, with 4.76 mg methyl ferulate as the control group and sterile deionized water as the blank control group; it was cultured in a constant temperature shaker at 28 °C. The sample was removed at specific intervals, 200 µL of culture medium was added to a sterile 96-well plate, the optical density (OD) value was measured at 595 nm, and 200 µL of sterile culture medium was added to maintain a constant total volume.

#### 2.3.2. Determination of Antioxidant Activity

Put 200 mg of *methyl ferulate/zein* fiber into 2 mL PBS (phosphate buffered saline), place it in an oscillator at room temperature and away from light, 120 rpm/min, and oscillate for 24 h. After the oscillation, take an appropriate amount of solution and verify the antioxidant activity of *methyl ferulate/zein* fiber membrane by using ABTS (2,2′-Azinobis-(3-ethylbenzthiazoline-6-sulphonate)) radical scavenging capacity test kit, DPPH (α,α-diphenyl-β-pricrylhydrazyl) radical scavenging capacity test kit and hydroxyl radical scavenging capacity test kit. The theoretical content of methyl ferulate in fiber membrane was 8.52 mg, with 8.52 mg methyl ferulate as the control group and the same amount of zein fiber as the blank group.

#### 2.3.3. Treatment and Grouping of Sea Bass

The sea bass (purchased from Linxi Street Aquatic Market of Jinzhou, Liaoning, China) was peeled, and the surface of the sea bass was washed with 0.85% normal saline, mucus; certain microorganisms were washed, and the muscle was cut into three groups on a significantly clean working table. The first group of bass tablets was coated with *methyl ferulate/zein* fiber membranes sterilized by ultraviolet light (15 min). The second group of bass slices was coated with a corn alcohol-soluble protein fiber membrane sterilized by ultraviolet light (15 min). The third group of uncoated bass slices (blank control group) was placed in sterile steaming bags and stored in a refrigerator at 4 °C for standby.

#### 2.3.4. Determination of TVC

The fish were placed on an ultra-clean worktable; 5 g of the fish sample was placed in a sterilized cooking bag consisting of 45 mL of 0.85% sterile normal saline, and patted for 120 s. The homogenate was diluted 10 times; 2–3 dilution multiples were selected, 1 mL was suctioned into the sterilized dish, which was followed by adding approximately 20 mL of sterilized plate counting agar medium and shaken well, and 3 parallel tests were conducted for each dilution. After the agar was coagulated, the plate was turned over and cultured in a 28 °C incubator for 48 h. Finally, the colonies were counted.

#### 2.3.5. Determination of pH

A total of 5 g of fish was weighed, ground and mixed; then, it was placed in a conical flask, and was diluted with sterile deionized water to a volume of 45 mL, left to stand for 30 min, and the pH value of the solution was measured with a pH meter.

#### 2.3.6. Determination of TVB-N Content

Five grams of minced fish was placed in a rectifying tube, which was followed by adding 25 mL of deionized water and 0.5 g of magnesium oxide powder in turn; the TVB-N of the fish fillets was determined using an automatic Kjeldahl nitrogen analyzer (KJELTEC 8400, Foss, Shanghai RuiFen International Trading Co., Ltd., Shanghai, China).

#### 2.3.7. Determination of TBA Content

Five grams of fish was weighed and ground into a conical flask, and 12.5 mL of deionized water was added, and the mixture was homogenized for 30 s. Then, 12.5 mL of 5% trichloroacetic acid was added and homogenized for 2 min, left for 30 min, and finally filtered. A total of 5 mL of filtrate was added to the colorimetric tube, and a 0.02 mol/L thiobarbituric acid solution was added to the tube to mix it evenly. The colorimetric tube remained at 80 °C for 40 min, cooled to room temperature, and the absorbance was measured at 532 nm using a UV-vis spectrophotometer. The TBA content is expressed as the malondialdehyde (mg) content in sea bass per kilogram.

### 2.4. Statistical Analysis

The results were obtained as the average value, and the analysis data were processed using Origin 2018 software, and the SPSS software (20.0) was used for statistical analysis. Statistical significance of mechanical properties test, release in vitro, antibacterial activity, microbiological analysis pH and TVB-N was set at *p* < 0.05, and with differences being insignificant at *p* > 0.05.

## 3. Results and Discussion

### 3.1. Characterization of the Electrospun Fibers

#### 3.1.1. Morphology Characterization

Fiber diameter plays a key role in the final performance of the electrospun fiber network and the properties of the composite and structure of the fibers prepared from it [20]. The surfaces of both the corn alcohol-soluble protein and *methyl ferulate/zein* nanofiber membrane was smooth and even; it was relatively easy to separate the fiber membrane from the aluminum foil. Figure 2A presents the SEM images of the nanofibers. The fibers prepared by electrospinning technology are random in direction, forming a network structure as well as a smooth surface without beads. The average diameter of the zein fibers was 185 nm, and the average diameter of the coaxial *methyl ferulate/zein* fiber was 322 nm. The increase in the diameter of the coaxial fiber may be due to the successful encapsulation of methyl ferulate in the fiber by zein. From the TEM image (Figure 2C), it could be seen that in the coaxial electrospinning process, *methyl ferulate/zein* formed a core–shell structure, and methyl ferulate was completely wrapped by zein. The results indicated that the average diameter of the coaxial nanofibers was larger than that of the uniaxial corn alcohol-soluble protein nanofibers due to the formation of the core–shell structure of *methyl ferulate/zein*. The core–shell structure is a remarkable feature of coaxial electrospinning; the active components can be effectively encapsulated in the shell. In medicine, coaxial electrospinning is often used as a drug carrier to improve drug stability and control drug release [21].

#### 3.1.2. Mechanical Properties Test

The mechanical properties of zein and *methyl ferulate/zein* films were tested, as shown in Table 1, as well as the elongation (σt) at the breaking point (εt). In the experiment, the electrospun fiber was deformed with an increase in the tensile force, and the fiber broke when the tensile force increased to a certain extent. The experimental data indicated that the zein fiber membrane had a sufficient tensile strength, which was similar to the mechanical properties of the electrospun fiber membrane of the zeol-soluble protein prepared by Wen et al. [22]. The tensile strength of the *methyl ferulate/zein* fiber membrane was significantly higher than that of the corn alcohol-soluble protein fiber membrane (*p* < 0.05), and the elongation at the breaking point was insignificant (*p* > 0.05). This indicates that the tensile strength of the *methyl ferulate/**ze**in* fiber membrane is sufficient, and the addition of methyl ferulate made the corn alcohol-soluble protein fiber membrane difficult to break. This may be because the phenolic acid structure of methyl ferulate increases the cross-linking between zein and methyl ferulate, makes the membrane flexible and improves the mechanical properties of the membrane [23], which is more suitable for food preservation.

#### 3.1.3. XRD Analysis

The XRD patterns of the *methyl ferulate/zein* fiber membrane, zein fiber membrane, and methyl ferulate powder are shown in Figure 3A. The main diffraction peaks of methyl ferulate powder appear at 2θ = 8.6°, 11.1°, 15.2°, 17.0°, 25.1°, 26.5°, and 27.4° (Figure 3A), which is a special diffraction peak of methyl ferulate, which indicates that the powder has a crystal structure. The wide diffraction peak of the zein fiber membrane appears at 2θ = 8.9° and 19.6° (Figure 3B), which is the unique diffraction peak of zein. The diffraction peak of the *methyl ferulate/zein* fiber membrane appeared at 2θ = 8.4° and 21.3° (Figure 3C), the sharpness of the peak decreased, and the width of the peak increased compared to that of zein. The crystal structure and orientation of biopolymer chains depend greatly on polymer properties (molecular weight, regularity and glass transition temperature), solvent medium and process variables [24]. The diffraction peak of the *methyl ferulate/zein* fiber membrane deviated slightly, which may be caused by the higher voltage applied in electrospinning, which resulted in a change in the electric field. The results indicated that the crystal properties of zein remained unchanged by the addition of methyl ferulate. However, the sharpness of the main diffraction peaks of methyl ferulate decreased or disappeared, indicating that the crystallinity of methyl ferulate decreased after encapsulation by zein.

#### 3.1.4. WCA Analysis

The measurement of the WCA can be used to express the hydrophobic/hydrophilic properties of the membrane, which is crucial for biological applications such as cell adhesion and active substance diffusion [25]. It can be seen from Figure 3B that the zein fiber membrane is hydrophobic, the WCA of the *methyl ferulate/zein* fiber membrane decreased to 54.85°, indicating that the fiber membrane was hydrophilic. The addition of methyl ferulate increased the hydrophilicity of the fiber membrane, indicating that the fiber membrane could better contact water and other substrates, which was helpful for the release of methyl ferulate. In addition, the size distribution and surface morphology of the electrospun fibers affect the hydrophilicity [26]. The addition of methyl ferulate increases the fiber diameter of the fiber membrane, which may limit the air retention at the fiber membrane water interface, which leads to an increase in hydrophilicity. Yao et al. [27] prepared a coaxial orange essential oil (OEO)/zein fiber membrane and found that the addition of OEO increased the hydrophilicity of the fiber membrane, which was similar to the experimental conclusion.

#### 3.1.5. Thermal Performance Analysis

The DSC and TG test results of the electrospun fibers are shown in Figure 4. It can be seen from Figure 4A that the glass transition temperature (Tg) of the film slightly decreased following the addition of methyl ferulate. This may be caused by the low Tg of methyl ferulate; the decrease in Tg in the fiber membrane was insignificant owing to the low content of methyl ferulate in the membrane. According to the thermogravimetry diagram in Figure 4B, the sample begins to decompose with increasing temperature. At approximately 100 °C, the samples were decomposed into small parts owing to the evaporation of the free and combined water in the sample. The TG curve of the sample tended to be stable at 160 °C and 250 °C, where the sample was no longer decomposed, and the weight loss rate was 98.4%. The pyrolysis of the zein fiber membrane was divided into the following two parts: the first part was in the range of 240–400 °C, and the weight loss rate was 55.1%, whereas the second part appeared in the range of 430–627 °C, and the weight loss rate was 35.5%. This is consistent with the research results of Xiao et al. [28], the thermal stability of pure gliadin nanofibers is higher than that of all composite fibers. The first part appeared in the range of 200–380 °C, the weight loss rate was 64.1%, and the second part appeared in the range of 425–630 °C and the weight loss rate was 25.5%. The weight loss rate of the electrospun fibers increased in the first part after methyl ferulate was added; however, the weight loss rate in the second part decreased. This indicates that the thermal stability of the electrospun fibers changed with the addition of methyl ferulate. The results demonstrated that the thermal stability of methyl ferulate could be improved by using a coaxial methyl ferulate/zeaxyl alcohol-soluble protein fiber membrane.

#### 3.1.6. FTIR Spectra

As shown in Figure 4C, the chemical structures of methyl ferulate and zein were analyzed by FTIR. The characteristic absorption peaks of zein are at 1650 cm^−1^ (-C=O stretching vibration) and 1545 cm^−1^ (N-H bending vibration) [29]. The absorption peaks did not disappear and there was no significant shift between the membrane of the corn alcohol-soluble protein and *methyl ferulate/zein*. The structure of zein did not change after the film formation. The characteristic absorption peaks of methyl ferulate are at 1702, 1607, and 1518 cm^−1^, indicating the -C=O stretching vibration, and 3406 cm^−1^ is related to the O-H expansion vibration. In the membrane of *methyl ferulate/zein*, the peak strength of methyl ferulate apparently decreased owing to the low content of methyl ferulate. The results demonstrate that the addition of methyl ferulate had no effect on the structure of zein, and the main physical binding between methyl ferulate and zeaxyl protein was found.

#### 3.1.7. Release in Vitro

It can be seen from Figure 4D that the in vitro release performance of coaxial *methyl ferulate/zein* fiber membrane. The release rate of free methyl ferulate was rapid at the initial stage, with a release rate of 85.3% at 12 h and 92.6% at 24 h. The *methyl ferulate/zein* fiber membrane was rapidly released in 0–8 h, and the release amount was 30.3% at 8 h. The release rate was 77.5% at 84 h. The release of 84–132 h gradually slowed and stabilized to 83.3% at 132 h. The results demonstrate that the coaxial methyl ferulate/zeaxyl protein fiber membrane had a sustained-release effect, and the release rate of the membrane was significantly different from that of the free methyl ferulate (*p* < 0.05). The coaxial electrospun fiber has an effective impact on the sustained biological activity of the methyl ferulate.

### 3.2. Coating Effects of the Electrospun Fibers on the Quality of Sea Bass during Cold Storage

#### 3.2.1. Verification of Antibacterial Activity

It is critical to verify the antibacterial activity of methyl ferulate in the coaxial methyl ferulate/zeaxyl alcohol-soluble protein fiber membrane to the superior corrupt bacteria (*S**. putrefaciens*) [30] in sea bass. Figure 5 shows the absorbance curve of *S. putrefaciens* suspension. The absorbance curves of *methyl ferulate/zein* fiber membrane group and methyl ferulate group were significantly lower than those of zein and zein fiber membrane group (*p* < 0.05). The growth rate of bacteria in 0–12 h was rapidly inhibited by soaking the *methyl ferulate/zein* fiber membrane in the culture medium of *S**. putrefaciens*; the absorbance of the suspension and the total number of bacteria decreased continuously from to 12–72 h. There was no significant difference between zein fiber membrane group and zein group (*p* > 0.05), indicating that pure zein had no antibacterial effect. The absorbance of the *methyl ferulate/zein* soluble protein fiber membrane group was higher than that of the methyl ferulate group at 0–30 h. This may be due to the slow-release effect of the fiber membrane, which resulted in a low concentration of methyl ferulate in the solution, and the bacteriostatic effect was weaker than that of the methyl ferulate group. The absorbance of the *methyl ferulate/zein* fiber membrane group was slightly lower than that of the methyl ferulate group at 30–72 h, which may be caused by the instability of methyl ferulate in the solution for a long time; the release of methyl ferulate in the fiber membrane was still being released. The results indicate that methyl ferulate in the fiber membrane can exert antibacterial activity and has the effect of continuous inhibition.

#### 3.2.2. Verification of Antioxidant Activity

Methyl ferulate has a phenolic acid structure and strong free radical scavenging activity [31]. Table 2 shows the scavenging activities of fiber membrane on DPPH free radical, ABTS free radical and hydroxyl free radical. Coaxial *methyl ferulate/zein* fiber membrane has strong antioxidant activity, and the DPPH radical scavenging rate is the highest. Zein fiber membrane had no antioxidant activity. The results show that the antioxidant activity of *methyl ferulate/zein* fiber membrane was lower than that of free methyl ferulate. The results showed that the release of methyl ferulate in the fiber membrane exerted a strong antioxidant activity. It shows that it can delay the oxidation of food in the process of wrapping food.

#### 3.2.3. Microbiological Analyses

Changes in the total number of colonies can directly reflect the degree of fish spoilage owing to the action of microorganisms [32]. As shown in Figure 6A, with the extension of time, the TVC of sea bass meat in each group showed an upward trend. The TVC of the blank group increased the fastest, reaching 5.07 lg (CFU/mL) on the third day, exceeding the first-grade freshness (4 lg (CFU/mL)); it was 6.65 lg (CFU/mL) at 6 days, exceeding the secondary freshness (6 lg (CFU/mL)), which was significantly higher than that in *methyl ferulate/zein* fiber membrane treatment group (*p* < 0.05). The TVC in the zein fiber membrane group was lower than that in the blank group, which may be due to the blocking of gas exchange inside and outside the membrane and affected the growth of bacteria; however, there was no significant difference in TVC (*p* > 0.05). The TVC of the zein fiber membrane group was 4.96 lg (CFU/mL) and 6.48 lg (CFU/mL) on the 3rd and 6th day, respectively, which did not delay the spoilage of fish. During the entire cold storage period, the growth rate of TVC in the *methyl ferulate/zein* fiber membrane group was the slowest; while it was 4.86 lg (CFU/mL) when the storage time was 6 days, it was 7.07 lg (CFU/mL) on day 12. During the entire cold storage period, the *methyl ferulate/zein* fiber membrane group had a sustained inhibitory effect on the microorganisms in fish, which may be due to the slow-release effect of the fiber membrane. Methyl ferulate could be continuously released from the zein fiber membranes, which could inhibit the growth of microorganisms in fish for a long time. The results indicated that the coaxial *methyl ferulate/zein* fiber membrane could inhibit the growth and reproduction of the spoilage microorganisms in fish.

#### 3.2.4. Determination of pH

According to Figure 6B, the pH value of sea bass during refrigeration first descends and then rises, which is consistent with the change in trend of the pH during the refrigeration period of the related fish meat [33]. In the early stage of refrigeration, the glycogen in fish meat is decomposed into lactic acid and other acids under the action of anaerobic conditions and enzymes, and the anaerobic respiration of microorganisms produces acid substances, which leads to a decrease in the pH value. After 6 days of cold storage, the pH value of the blank group was significantly higher than that of the *methyl ferulate/zein* fiber membrane group (*p* < 0.05). The pH value of the white group was 6.86 on the 9th day of cold storage, 6.85 in the corn alcohol-soluble protein fiber membrane group; the increasing trend in pH was apparent. The pH value of the *methyl ferulate/zein* fiber membrane group was the slowest and most stable in the later stage. In 15 days, the pH value was 6.81, which was significantly lower than that of the first two groups (*p* < 0.05). The results indicate that the coaxial *methyl ferulate/zein* fiber membrane can effectively inhibit the growth of microorganisms in fish meat and slow down the production and accumulation of alkaline substances, thus controlling the increase in the pH value and maintaining freshness.

#### 3.2.5. Determination of TVB-N Content

TVB-N is an important indicator for determining the degree of fish spoilage, which indicates that fish proteins are decomposed by endogenous enzymes and microorganisms to produce volatile amines [34]. As shown in Figure 6C, the TVB-N content in each group gradually increased with the extension of cold storage time. In the early stage of cold storage, the protein was decomposed by proteases to produce amines and the TVB-N value of each group increased slowly. After 6 days of cold storage, the dominant spoilage bacteria, such as *S**. putrefaciens*, propagated in large numbers, producing a significant amount of amines such as ammonia, dimethylamine, and trimethylamine, which made the total viable counts and the pH value rise rapidly at this time and resulting in a rapid increase in the TVB-N value. The TVB-N value of the blank control group increased the fastest, which was significantly higher than that of the *methyl ferulate/zein* fiber membrane after 9 days of cold storage (*p* < 0.05). The TVB-N value of *methyl ferulate/zein* fiber membrane group presented the slowest rising trend, which was 18.04 mg/100 g at 12 d and 22.36 mg/100 g at 15 d. These results indicate that the *methyl ferulate/zein* fiber membrane can inhibit the growth of amine-producing spoilage bacteria such as *S**. putrefaciens*, delay the accumulation of amines, and thus slow down the growth of the TVB-N value.

#### 3.2.6. Determination of TBA Content

TBA is an important index for lipid oxidation and can be used to determine the degree of fatty oxidation and acid failure in aquatic products [35]. According to Figure 6D, the TBA content in fish meat increased with the increase of cold storage time. The TBA value of the blank control group increased most rapidly; the TBA value was 0.483 mg MDA kg^−1^ at 15 days. The TBA value of the zein fiber membrane group was slightly lower than that of the control group; however, the difference was insignificant (*p* > 0.05). The TBA value of the *methyl ferulate/zein* fiber membrane group increased the slowest; the TBA value was 0.365 mg MDA·kg^−1^ at 15 days. The results demonstrated that methyl ferulate had sufficient antioxidant properties and could effectively inhibit the growth and reproduction of microorganisms, as well as reduce the use rate of fat oxidation. Therefore, the TBA value of the *methyl ferulate/zein* fiber membrane group was lower than that of the blank control group. The results indicated that the coaxial *methyl ferulate/zein* fiber membrane can protect the oxidation of unsaturated fatty acids in sea bass and maintain its nutritional quality.

## 4. Conclusions

In this paper, a coaxial *methyl ferulate/zein* fiber membrane was successfully prepared by electrospinning. The SEM results showed that the fiber had irregular orientation, network structure, and the smooth surface without beads. The diameter of the fiber increased after methyl ferulate was added. TEM results indicated that the membrane was packed with methyl ferulate in the zein with the core–shell structure. The mechanical properties test showed that the addition of methyl ferulate increased the tensile strength of the zein fiber membrane and hardened the fiber membrane. X-ray diffraction confirmed that the crystallinity of methyl ferulate decreased after being encapsulated by the zeaxyl alcohol solution protein, and the crystallization ability of the polymer became weaker. The infrared spectrum indicated that the addition of methyl ferulate had no effect on the structure of zein, and the two were mainly bound physically. The thermal properties analysis showed that the coaxial *methyl ferulate/zein* fiber membrane could improve the thermal stability of methyl ferulate. The results demonstrated that the addition of methyl ferulate changed the membrane from hydrophobic to hydrophilic. The in vitro release demonstrated that the *methyl ferulate/zein* fiber membrane had the effect of sustained release. *Methyl ferulate/zein* fiber membranes have a sustained antibacterial effect on the *S. putrefaciens*. After the treatment of sea bass meat with *methyl ferulate/zein* fiber membrane, the fiber membrane can inhibit the growth of microorganisms in fish and effectively inhibit the increase in the pH value and TVB-N content of fish. In conclusion, methyl ferulate was successfully encapsulated in a corn alcohol-soluble protein fiber membrane, which has a fresh-keeping effect on sea bass and can be potentially used in food packaging.

## Figures and Tables

**Figure 1 foods-10-02385-f001:**
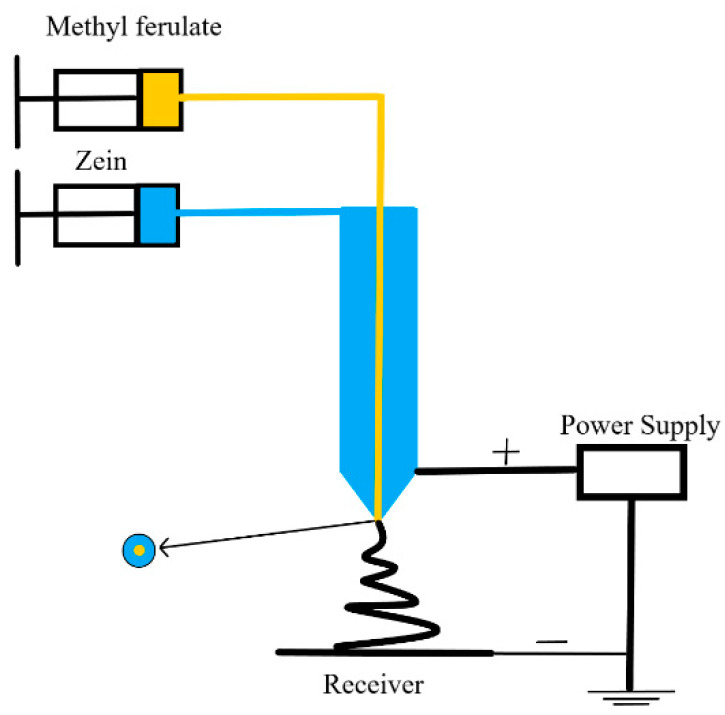
Schematic diagram of coaxial electrospinning device.

**Figure 2 foods-10-02385-f002:**
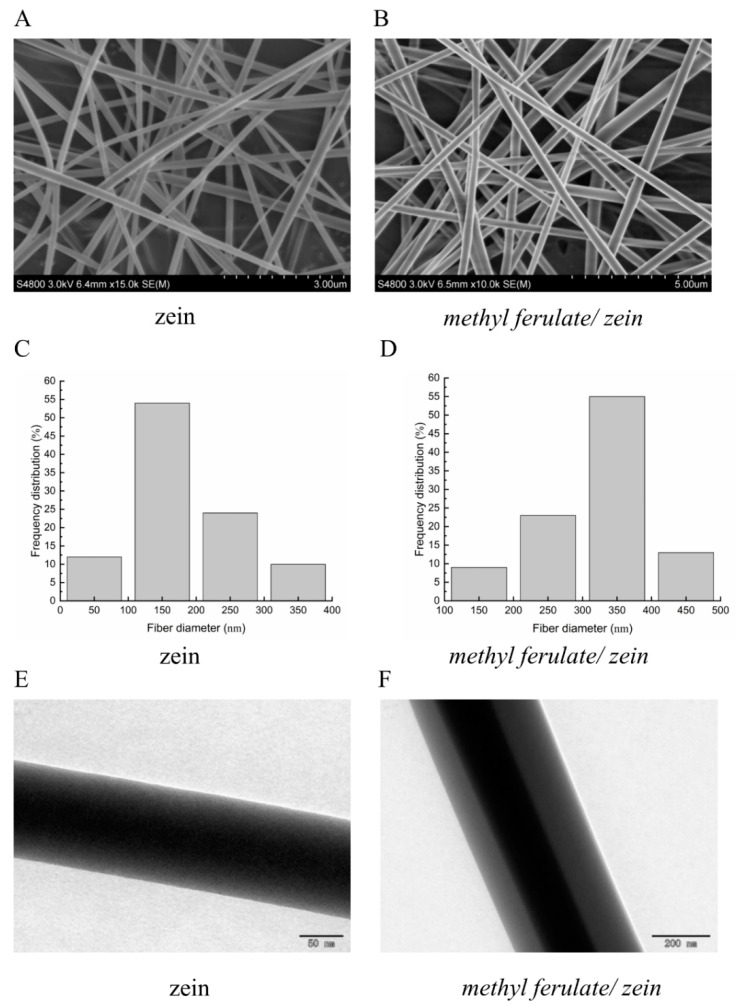
(**A**) SEM images of zein electrospinning fibers. (**B**) SEM images of *methyl ferulate/zein* electrospinning fibers. (**C**) Diameter distribution histogram of zein electrospinning fibers. (**D**) Diameter distribution histogram of *methyl ferulate/zein* electrospinning fibers. (**E**) TEM images of zein electrospinning fibers. (**F**) TEM images of *methyl ferulate/zein* fibers.

**Figure 3 foods-10-02385-f003:**
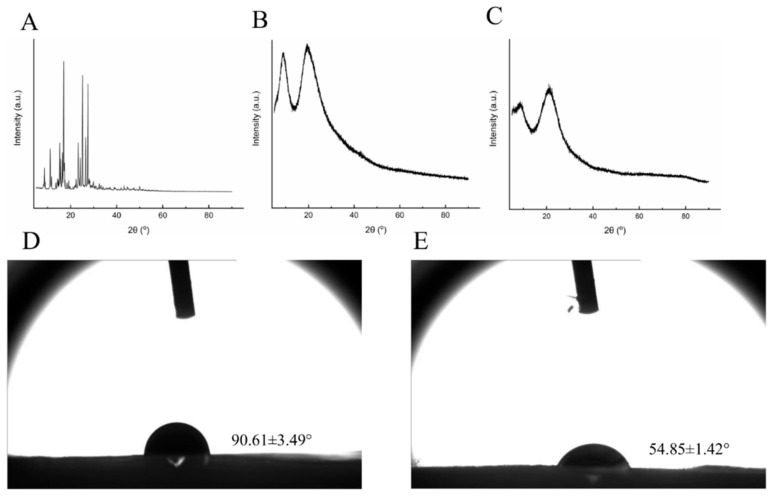
(**A**) X-ray diffraction patterns of methyl ferulate electrospinning fibers. (**B**) X-ray diffraction patterns of zein electrospinning fibers. (**C**) X-ray diffraction patterns of *methyl ferulate/zein* electrospinning fibers. (**D**) WCA of zein electrospinning fibers. (**E**) WCA of *methyl ferulate/zein* electrospinning fibers.

**Figure 4 foods-10-02385-f004:**
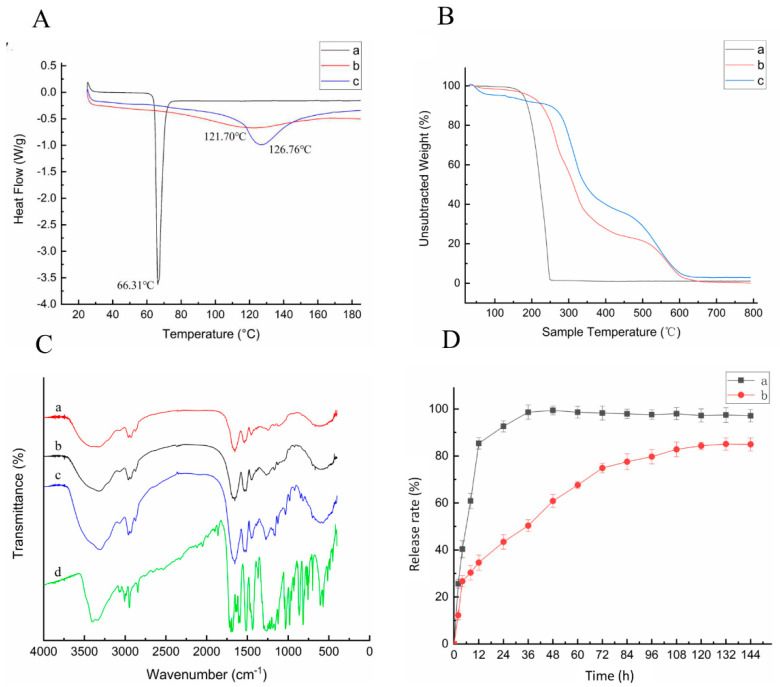
(**A**) DSC results of electrospinning fibers. (**B**) TG(thermogravimetric analysis) of electrospinning fibers (a–c respectively present methyl ferulate, *methyl ferulate/zein* fiber membrane, and zein fiber membrane). (**C**) FTIR(Fourier-transform infrared spectroscopy) spectra (a–d respectively present zein fiber membrane, zein, *methyl ferulate/zein* fiber membrane, methyl ferulate). (**D**) Slow-release properties of electrospinning fibers membranes (a, b respectively represents free methyl ferulate and *methyl ferulate/zein* fiber membrane).

**Figure 5 foods-10-02385-f005:**
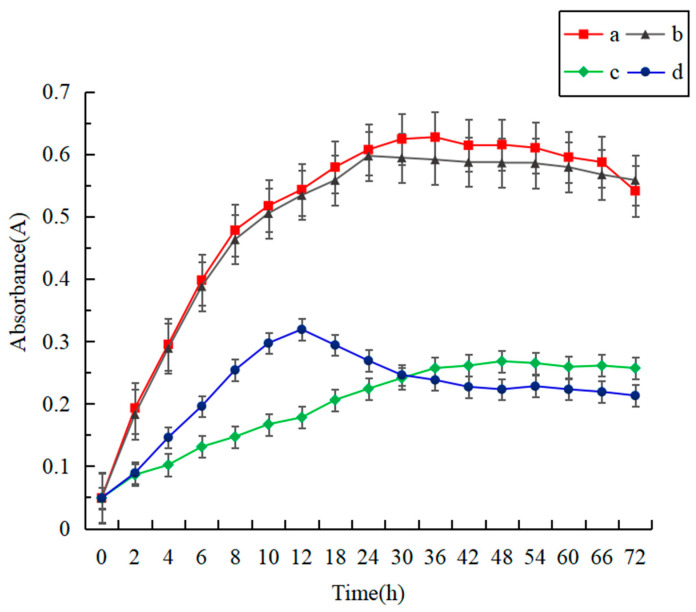
Verification of antibacterial activity of fiber membrane (a–d respectively represent zein, zein fiber membrane, free methyl ferulate, *methyl ferulate/zein* fiber membrane).

**Figure 6 foods-10-02385-f006:**
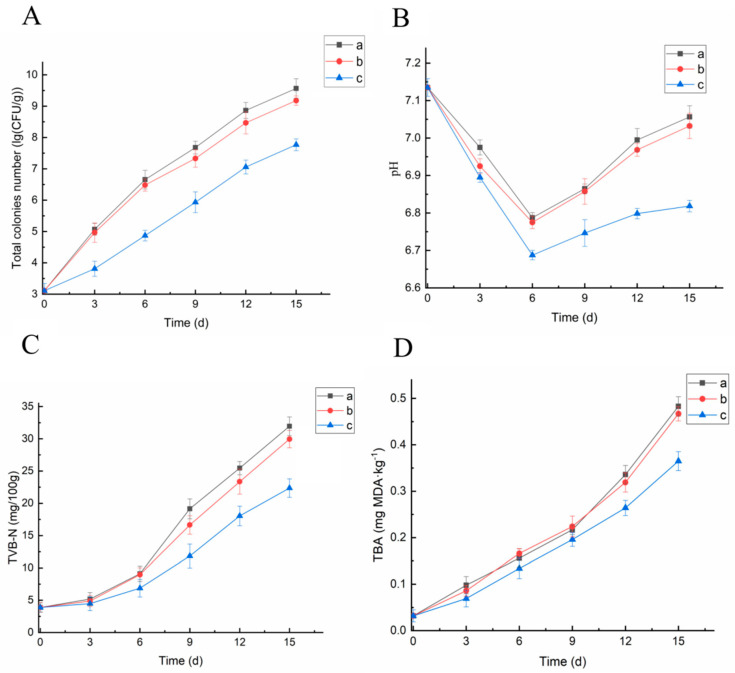
Preservation effect of methyl ferulate electrospun film on sea bass. (**A**) Changes of TVC. (**B**) pH change. (**C**) Changes of TVB-N. (**D**) Changes of TBA (a–c respectively represents blank control, zein fiber membrane, and *methyl ferulate/zein* fiber membrane).

**Table 1 foods-10-02385-t001:** Mechanical properties of fiber membrane.

Fiber Membrane	Tensile Strength/MPa	Elongation at Break/%
Zein	19.685 ± 0.445 ^b^	14.32 ± 1.64 ^a^
*Methyl ferulate/zein*	27.035 ± 0.561 ^a^	11.18 ± 1.07 ^a^

Note: ^a,b^ Significantly different means (*p* < 0.05).

**Table 2 foods-10-02385-t002:** Verification of antioxidant activity of fiber membrane.

Electrospun Membrane	DPPH Radical Scavenging Rate (%)	ABTS Radical Scavenging Rate (%)	Hydroxyl Radical Scavenging Rate (%)
Zein electrospun membrane	15.49 ± 1.58 ^c^	9.86 ± 1.23 ^c^	16.42 ± 1.69 ^c^
Free methyl ferulate	93.17 ± 4.86 ^a^	85.67 ± 4.95 ^a^	92.02 ± 3.52 ^a^
*M**ethyl ferulate/zein* electrospun membrane	79.65 ± 3.65 ^b^	73.43 ± 1.96 ^b^	71.82 ± 2.51 ^b^

Note: ^a–c^ Significantly different means (*p* < 0.05). DPPH: α, α-diphenyl-β-pricrylhydrazyl. ABTS: 2,2′-Azinobis-(3-ethylbenzthiazoline-6-sulphonate).

## Data Availability

Not applicable.

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
