# Peer review of "Antibacterial Properties of Coaxial Spinning Membrane of Methyl ferulate/zein and Its Preservation Effect on Sea Bass"

_foods, 2021, doi:10.3390/foods10102385_

Round 1

Reviewer 1 Report

This research article studies the inclusion of methyl ferulate in fiber membrane of zein obtained by electrospinning for the fish preservation. The developed fibers are characterized by determining their morphological, mechanical and thermal properties, and by XRD, FIR, WCA. In addition, the authors evaluate its antimicrobial and antioxidant properties.

In general, the manuscript is well written. The paper is well understood, and is easy and fast to read. The English language is understandable and correct. Title, author list and affiliations, abstract and keywords comply with the authors guide of the website (https://www.mdpi.com/journal/foods/instructions#referees). The materials and methods allow the results to be reproduced. The experiments fit a suitable design well. In general, plagiarism is not detected (https://plagiarismdetector.net/es).

However, to contribute to the improvement of the article, I propose some changes that are detailed below.

- Although the results are well presented, the discussion of each section is too brief. It would be good to present longer discussions and provide a bibliography for it.

- Unify spelling criteria. For example, coaxial is different in title and in line 374.

- The binomial scientific name (genus species) must be highlighted, at least, in italics (title, lines 482, 484, 485, 488, 558, 563-564, 571).

- There are missing or excess spaces, spelling errors… in lines: 17, 152, 172, 188, 198, 200, 282, 285, 322, 337, 346, 369, 373, 374, 374, 375, 376, 376, 377, 406, 407, 490, 519, 525, 526 (Part), and 526 (coumarate).

- Please, consistency in the units and format throughout the manuscript. For example: mL or ml (lines 101, 102, 103 vs. 94, 98, 103 and 104), Figure (line 101) vs. Fig (in the rest of the manuscript).

- In line 101, Figure 4-1 is named, which is not in the manuscript.

- In lines 114, 141,143 and 156 ... Which country do you refer to?

- Check the capitalization of the section titles (line 159) and table 2.

- The header of Table 1 and Table 2 have different format. Missing "Fig 4C" in line 326. what is "d" in figure 4C? In Figure 4D, "a" is methyl ferulate?

- Consistency with the rest of the manuscript in the use of the first or last name in line 256 vs. 541, and in reference 1, 24, 26...

- In line 226 ... should it read "Results and Discussion"?

- It is recommended not to repeat the data in the text and in the figures (line 289-291).

- In figure 5 the growth curve of a control is not observed; therefore, it is not possible to know if zein fiber membrane has an effect or not on the growth of Shewanella putrefaciens.

- The text in lines 392-394 is not easily understood.

- What do you mean by the term " shivaia putrefactory " in line 468? It is not a common concept. Please describe what it is.

- In general, the references used in the manuscript could be more specific and adequate.

- Please check the bibliography throughout the manuscript. For example, [7] does not have the information cited in lines 39-42, that information is found in reference [8]. In reference 14, the formulas described are not found either. Reference 22 appears in the manuscript after reference 23.

- References should be described (Journal Articles): 1. Author 1, A.B.; Author 2, C.D. Title of the article. Abbreviated Journal Name YearVolume, page range. Please change the format of the references.

In reference 1, an author is missing. In reference 17, 18, 21, 22, 23, 24, 26, 27, 28 and 29 are missing two authors. And in reference 14 and 16 are missing four authors. In other references, you have named up to seven authors. Please unify the criteria.

In reference 2, the DOI is separate from the rest of the reference. Please join it.

In reference 14, the name of the journal is not abbreviated (Bioorg Med Chem Lett.). In reference 29, the name of the journal is not abbreviated (Int. J. Biol. Macromol.).

Reference 15 is wrong, it should be: [15] Kikuzaki, H., Hisamoto, M., Hirose, K., Akiyama, K., & Taniguchi, H. Antioxidant properties of ferulic acid and its related compounds. J. Agric. Food Chem. 2002, 50(7), 2161-2168. Reference 30 is wrong, it should be: [30] Luo, Y.

Author Response

Dear Reviewers:

Thanks very much for taking your time to review this manuscript. I really appreciate all your comments and suggestions! Please find my itemized responses in below and my revisions in the re-submitted files.

Thanks again!

Review 1

Comment 1: Although the results are well presented, the discussion of each section is too brief. It would be good to present longer discussions and provide a bibliography for it.

Response: We increase the discussions of Morphology characterization (Line 311-314,page 11), Mechanical properties test (Line 328-330, page11), XRD analysis (Line 345-348, page 12), Thermal performance analysis (Line 384-386, page13), and Verification of antioxidant activity (Line451-452, page 15),and add the corresponding references.([21], [23],[24],[28]).

Comment 2: Unify spelling criteria. For example, coaxial is different in title and in line 374.

Response: We have already mofidied the “Co axial” to “Coaxial” in line 446.

Comment 3: The binomial scientific name (genus species) must be highlighted, at least, in italics (title, lines 482, 484, 485, 488, 558, 563-564, 571).

Response: We have mofidied the “methyl ferulate/zein” to “methyl ferulate/zein” (title, lines 44, 110, 463, 509,and so on ) and the name of genus species “Shigella putrefaciens ” to “S. putrefaciens in line 424,429.

Comment 4: There are missing or excess spaces, spelling errors… in lines: 17, 152, 172, 188, 198, 200, 282, 285, 322, 337, 346, 369, 373, 374, 374, 375, 376, 376, 377, 406, 407, 490, 519, 525, 526 (Part), and 526 (coumarate).

Response: We have mofidied the excess spaces and spelling errors in the full text.

Comment 5: Please, consistency in the units and format throughout the manuscript. For example: mL or ml (lines 101, 102, 103 vs. 94, 98, 103 and 104), Figure (line 101) vs. Fig (in the rest of the manuscript).

Response: We have uniformly changed “ml” into “mL” (Lines 150-152, 221, 243, 282....... ), and changed “Figure” to “Fig” (Line150).

Comment 6: In line 101, Figure 4-1 is named, which is not in the manuscript.

Response: This is our mistake, Figure 4-1 refers to Fig. 1.

Comment 7: In lines 114, 141,143 and 156 ... Which country do you refer to?

Response: We are so sorry about the lines you mean , but we guess you may doubt about the expression ,so we changed the expression of lines 129, 159, 165,191,209...

Comment 8:  Check the capitalization of the section titles (line 159) and table 2

Response: We have mofidied the section titles “Characterization of the Electrospun Fibers” to “Characterization of the electrospun fibers”(line 134) and Table 2(line 455).

Comment 9: The header of Table 1 and Table 2 have different format. Missing "Fig 4C" in line 326. what is "d" in figure 4C? In Figure 4D, "a" is methyl ferulate?

Response :We have unified the header of format of Table 1(line 333) and Table 2 (line454), and changed “Fig 4C” to “Fig. 4C”.The “d” in Fig. 4C is methyl ferulate. In Fig. 4D, a and b are respectively present free methyl ferulate and methyl ferulate/zein fiber membrane.

Comment 10: Consistency with the rest of the manuscript in the use of the first or last name in line 256 vs. 541, and in reference 1, 24, 26...

Response: We have changed the names of the manuscript to those in the references.(Lines 367 and the references)

Comment 11: In line 226 ... should it read "Results and Discussion"?

Response: We have changed in line 293.

Comment 12: It is recommended not to repeat the data in the text and in the figures (line 289-291).

Response : We have deleted duplicate data in the text (lines 358-360).

Comment 13: In figure 5 the growth curve of a control is not observed; therefore, it is not possible to know if zein fiber membrane has an effect or not on the growth of Shewanella putrefaciens.

Response :We filled up the missing data.

Fig. 5 Verification of antibacterial activity of fiber membrane. a, b, c, d are respectively present zein, zein fiber membrane, free methyl ferulate, methyl ferulate/zein fiber membrane.

Comment 14: The text in lines 392-394 is not easily understood.

Response: These sentences are an explanation of Fig. 6A. We changed “With the extension of time, the TVC of sea bass meat in each group showed an upward trend. The TVC of the blank group increased the fastest, reaching 5.07 lg (CFU/mL) on the third day, exceeding the first grade freshness (4 lg (CFU/mL)); It was 6.65 lg (CFU / mL) at 6 days, which exceeded the secondary freshness (6 lg (CFU/mL)), which was significantly higher than that in methyl ferulate/zein fiber membrane treatment group (p < 0.05)”

Comment 15: What do you mean by the term " shivaia putrefactory " in line 468? It is not a common concept. Please describe what it is.

Response: This is a spelling mistake.We have changed it to “S. putrefaciens”(line 548).

Comment 16: In general, the references used in the manuscript could be more specific and adequate.

Response: We added some references (21, 23, 24, 25, 26, 28, 31) and made the content of the article as specific as possible.

Comment 17: Please check the bibliography throughout the manuscript. For example, [7] does not have the information cited in lines 39-42, that information is found in reference [8]. In reference 14, the formulas described are not found either. Reference 22 appears in the manuscript after reference 23.

Response: We have replaced reference [7] with “Keyur, Desai, Kevin, et al. Morphological and Surface Properties of Electrospun Chitosan Nanofibers[J]. Macromolecules, 2008, 9(3):1000–1006.” The formulas in reference 19,we have changed (line 178).And we corrected the order of references 22 and 23 (line 322, 365).

Comment 18: References should be described (Journal Articles): 1. Author 1, A.B.; Author 2, C.D. Title of the article. Abbreviated Journal Name Year, Volume, page range. Please change the format of the references.

Response: We have corrected and unified the format of references (line 566-671).

Comment 19: In reference 1, an author is missing. In reference 17, 18, 21, 22, 23, 24, 26, 27, 28 and 29 are missing two authors. And in reference 14 and 16 are missing four authors. In other references, you have named up to seven authors. Please unify the criteria.

Response: We have unified the criteria of authors in the reference,only 3 authors appeared in all references.

Comment 20: In reference 2, the DOI is separate from the rest of the reference. Please join it.

Response: We deleted DOI from all references.

Comment 21: In reference 14, the name of the journal is not abbreviated (Bioorg Med Chem Lett.). In reference 29, the name of the journal is not abbreviated (Int. J. Biol. Macromol.).

Response:We unified the name of the journal,wrote their full names.

Comment 22: Reference 15 is wrong, it should be: [15] Kikuzaki, H., Hisamoto, M., Hirose, K., Akiyama, K., & Taniguchi, H. Antioxidant properties of ferulic acid and its related compounds. J. Agric. Food Chem. 2002, 50(7), 2161-2168. Reference 30 is wrong, it should be: [30] Luo, Y.

Response: We have modified the format of references 15 and 330(lines 608, 669).

Review 2

Comment 1: Lines 183-189: These 2 sentences should be grammatically corrected

Response: We have corrected these 2 sentences. “Put 200 mg of methyl ferulate/zein fiber into 2 mL PBS, place it in an oscillator at room temperature and away from light, 120 rpm/ min, and oscillate for 24 h. After the oscillation, take an appropriate amount of solution and verify the antioxidant activity of methyl ferulate/zein fiber membrane by using ABTS radical scavenging capacity test kit, DPPH radical scavenging capacity test kit and hydroxyl radical scavenging capacity test kit. The theoretical content of methyl ferulate in fiber membrane was 8.52 mg, with 8.52 mg methyl ferulate as the control group and the same amount of zein fiber as the blank group. (Lines 243-250)  

Comment 2: Regarding statistical analysis, it would be worth adding what tests were used and what was the factor studied.

Response: We added the tests were used, including mechanical properties test, release in vitro, antibacterial activity, microbiological analyses pH and TVB-N (lines 291,292)

Comment 3:Line 159, 222: These subsection titles are uppercase, others lowercase? Check editorial requirements.

Response: We have changed the case of subtitles(line216, 238).

Comment 4:  I would suggest to change the form of sentences so that they do not start like this, for example: “Fig. 3A presents the XRD ..” (Line 267) and also Lines 183, 2015, 218,

Response: We have revised this statement (lines 339, 271, 275).

Comment 5: Line 486: https://doi.org/10.2377/0003-925X-58-98. - correct it

Response: We deleted DOI from all references and modified the format of references.

Comment 6: Also, check the entire manuscript for editing, sometimes spaces are missing, sometimes there is an excess of them.

Response: We checked the manuscript and deleted the extra space.

Reviewer 2 Report

The manuscript is well prepared in terms of content, it is scientifically valuable, and the proposed coatings with methyl ferulate that was embedded into zein by electrospinning technology to prepare fiber membrane may find the wider application on an industrial scale. The manuscript includes a series of necessary tests that were used to evaluate the application of the new coating. It meets the current need because it creates the possibility of extending the storage of a specific product, but perhaps such a solution can be used for a different type of food. Also important, methyl ferulate is a natural antibacterial agent with strong activity and low toxicity. This is very important because consumers are now looking for food from the so-called clean labels, i.e. without the addition of artificially manufactured substances. However, the authors should take into account the toxicological risks. Perhaps it will be included in their subsequent research.

The manuscript is well written and organized, there are no comments on the title, abstract, methodology result, and discussion and conclusions. However, many minor points may not be relevant to editorial requirements unless they have just been updated, so I suggest you check them out. For example, the structure of the manuscript in terms of the order of the main chapters (applies to "Materials and methods"), the way of signing figures, and especially the preparation of "References". The literature contains 30 important sources, including 9 from the last 5 years, all closely related to the subject of the manuscript.

 Some other comments:

 Lines 183-189: These 2 sentences should be grammatically corrected:

“200 mg of methyl ferulate/zeaxyl protein fiber was put into 2 ml PBS, and the theoretical content of methyl ferulate in the fiber membrane was 8.52 mg, 8.52 mg of methyl ferulate was taken as control group, and the same amount of corn alcohol-soluble protein fiber was used as the blank group. Then take appropriate solution, and use ABTS free radical scavenging ability detection kit, DPPH free radical scavenging ability detection kit and hydroxyl free radical scavenging ability test kit to verify the antioxidant activity of methyl ferulate / zein fiber membrane.”

Regarding statistical analysis, it would be worth adding what tests were used and what was the factor studied.

Line 159, 222: These subsection titles are uppercase, others lowercase? Check editorial requirements.

I would suggest to change the form of sentences so that they do not start like this, for example:

“Fig. 3A presents the XRD ..” (Line 267) and also Lines 183, 2015, 218,

Line 486: https://doi.org/10.2377/0003-925X-58-98. - correct it

Also, check the entire manuscript for editing, sometimes spaces are missing, sometimes there is an excess of them..

Author Response

Dear Reviewers:

Thanks very much for taking your time to review this manuscript. I really appreciate all your comments and suggestions! Please find my itemized responses in below and my revisions in the re-submitted files.

Thanks again!

Review 1

Comment 1: Although the results are well presented, the discussion of each section is too brief. It would be good to present longer discussions and provide a bibliography for it.

Response: We increase the discussions of Morphology characterization (Line 311-314,page 11), Mechanical properties test (Line 328-330, page 11), XRD analysis (Line 345-348, page 12), Thermal performance analysis (Line 384-386, page 13), and Verification of antioxidant activity (Line 451-452, page 15),and add the corresponding references.([21], [23],[24],[28]).

Comment 2: Unify spelling criteria. For example, coaxial is different in title and in line 374.

Response: We have already modified the “Co axial” to “Coaxial” in line 446.

Comment 3: The binomial scientific name (genus species) must be highlighted, at least, in italics (title, lines 482, 484, 485, 488, 558, 563-564, 571).

Response: We have modified the “methyl ferulate/zein” to “methyl ferulate/zein” (title, lines 44, 110, 463, 509,and so on ) and the name of genus species “Shigella putrefaciens ” to “S. putrefaciens in line 424,429.

Comment 4: There are missing or excess spaces, spelling errors… in lines: 17, 152, 172, 188, 198, 200, 282, 285, 322, 337, 346, 369, 373, 374, 374, 375, 376, 376, 377, 406, 407, 490, 519, 525, 526 (Part), and 526 (coumarate).

Response: We have modified the excess spaces and spelling errors in the full text.

Comment 5: Please, consistency in the units and format throughout the manuscript. For example: mL or ml (lines 101, 102, 103 vs. 94, 98, 103 and 104), Figure (line 101) vs. Fig (in the rest of the manuscript).

Response: We have uniformly changed “ml” into “mL” (Lines 150-152, 221, 243, 282....... ), and changed “Figure” to “Fig” (Line 150).

Comment 6: In line 101, Figure 4-1 is named, which is not in the manuscript.

Response: This is our mistake, Figure 4-1 refers to Fig. 1.

Comment 7: In lines 114, 141,143 and 156 ... Which country do you refer to?

Response: We are so sorry about the lines you mean , but we guess you may doubt about the expression ,so we changed the expression of lines 129, 159, 165,191,209...

Comment 8:  Check the capitalization of the section titles (line 159) and table 2

Response: We have modified the section titles “Characterization of the Electrospun Fibers” to “Characterization of the electrospun fibers”(line 134) and Table 2(line 455).

Comment 9: The header of Table 1 and Table 2 have different format. Missing "Fig 4C" in line 326. what is "d" in figure 4C? In Figure 4D, "a" is methyl ferulate?

Response :We have unified the header of format of Table 1(line 333) and Table 2 (line 454), and changed “Fig 4C” to “Fig. 4C”.The “d” in Fig. 4C is methyl ferulate. In Fig. 4D, a and b are respectively present free methyl ferulate and methyl ferulate/zein fiber membrane.

Comment 10: Consistency with the rest of the manuscript in the use of the first or last name in line 256 vs. 541, and in reference 1, 24, 26...

Response: We have changed the names of the manuscript to those in the references.(Lines 367 and the references)

Comment 11: In line 226 ... should it read "Results and Discussion"?

Response: We have changed in line 293.

Comment 12: It is recommended not to repeat the data in the text and in the figures (line 289-291).

Response : We have deleted duplicate data in the text (lines 358-360).

Comment 13: In figure 5 the growth curve of a control is not observed; therefore, it is not possible to know if zein fiber membrane has an effect or not on the growth of Shewanella putrefaciens.

Response :We filled up the missing data.

Fig. 5 Verification of antibacterial activity of fiber membrane. a, b, c, d are respectively present zein, zein fiber membrane, free methyl ferulate, methyl ferulate/zein fiber membrane.

Comment 14: The text in lines 392-394 is not easily understood.

Response: These sentences are an explanation of Fig. 6A. We changed “With the extension of time, the TVC of sea bass meat in each group showed an upward trend. The TVC of the blank group increased the fastest, reaching 5.07 lg (CFU/mL) on the third day, exceeding the first grade freshness (4 lg (CFU/mL)); It was 6.65 lg (CFU /mL) at 6 days, which exceeded the secondary freshness (6 lg (CFU/mL)), which was significantly higher than that in methyl ferulate/zein fiber membrane treatment group (p < 0.05)”

Comment 15: What do you mean by the term " shivaia putrefactory " in line 468? It is not a common concept. Please describe what it is.

Response: This is a spelling mistake.We have changed it to “S. putrefaciens”(line 548).

Comment 16: In general, the references used in the manuscript could be more specific and adequate.

Response: We added some references (21, 23, 24, 25, 26, 28, 31) and made the content of the article as specific as possible.

Comment 17: Please check the bibliography throughout the manuscript. For example, [7] does not have the information cited in lines 39-42, that information is found in reference [8]. In reference 14, the formulas described are not found either. Reference 22 appears in the manuscript after reference 23.

Response: We have replaced reference [7] with “Keyur, Desai, Kevin, et al. Morphological and Surface Properties of Electrospun Chitosan Nanofibers[J]. Macromolecules, 2008, 9(3):1000–1006.” The formulas in reference 19,we have changed (line 178).And we corrected the order of references 22 and 23 (line 322, 365).

Comment 18: References should be described (Journal Articles): 1. Author 1, A.B.; Author 2, C.D. Title of the article. Abbreviated Journal Name Year, Volume, page range. Please change the format of the references.

Response: We have corrected and unified the format of references (line 566-671).

Comment 19: In reference 1, an author is missing. In reference 17, 18, 21, 22, 23, 24, 26, 27, 28 and 29 are missing two authors. And in reference 14 and 16 are missing four authors. In other references, you have named up to seven authors. Please unify the criteria.

Response: We have unified the criteria of authors in the reference,only 3 authors appeared in all references.

Comment 20: In reference 2, the DOI is separate from the rest of the reference. Please join it.

Response: We deleted DOI from all references.

Comment 21: In reference 14, the name of the journal is not abbreviated (Bioorg Med Chem Lett.). In reference 29, the name of the journal is not abbreviated (Int. J. Biol. Macromol.).

Response:We unified the name of the journal,wrote their full names.

Comment 22: Reference 15 is wrong, it should be: [15] Kikuzaki, H., Hisamoto, M., Hirose, K., Akiyama, K., & Taniguchi, H. Antioxidant properties of ferulic acid and its related compounds. J. Agric. Food Chem. 2002, 50(7), 2161-2168. Reference 30 is wrong, it should be: [30] Luo, Y.

Response: We have modified the format of references 15 and 330(lines 608, 669).

Reviewer 2

Comment 1: Lines 183-189: These 2 sentences should be grammatically corrected

Response: We have corrected these 2 sentences. “Put 200 mg of methyl ferulate/zein fiber into 2 mL PBS, place it in an oscillator at room temperature and away from light, 120 rpm/ min, and oscillate for 24 h. After the oscillation, take an appropriate amount of solution and verify the antioxidant activity of methyl ferulate/zein fiber membrane by using ABTS radical scavenging capacity test kit, DPPH radical scavenging capacity test kit and hydroxyl radical scavenging capacity test kit. The theoretical content of methyl ferulate in fiber membrane was 8.52 mg, with 8.52 mg methyl ferulate as the control group and the same amount of zein fiber as the blank group. (Lines 243-250)  

Comment 2: Regarding statistical analysis, it would be worth adding what tests were used and what was the factor studied.

Response: We added the tests were used, including mechanical properties test, release in vitro, antibacterial activity, microbiological analyses pH and TVB-N (lines 291,292)

Comment 3:Line 159, 222: These subsection titles are uppercase, others lowercase? Check editorial requirements.

Response: We have changed the case of subtitles(line 216, 238).

Comment 4:  I would suggest to change the form of sentences so that they do not start like this, for example: “Fig. 3A presents the XRD ..” (Line 267) and also Lines 183, 2015, 218,

Response: We have revised this statement (lines 339, 271, 275).

Comment 5: Line 486: https://doi.org/10.2377/0003-925X-58-98. - correct it

Response: We deleted DOI from all references and modified the format of references.

Comment 6: Also, check the entire manuscript for editing, sometimes spaces are missing, sometimes there is an excess of them.

Response: We checked the manuscript and deleted the extra space.

Round 2

Reviewer 1 Report

Thank you for making such a thorough correction.

However, there are still formatting errors (especially in the figure captions), and, in the reference list, the journals are not abbreviated, the binomial scientific name are not italicized, and [J] appears in each reference.

Also, it would be good if the superscript letters in the tables were explained in the table´s footnotes or figure caption.

Otherwise ... good job!

Author Response

Dear Reviewer:

Thanks very much for taking your time to review our revised manuscript. I really appreciate all your comments and suggestions! Please find my itemized responses in below and my revisions in the re-submitted files.

Thanks again!

Comment 1: However, there are still formatting errors (especially in the figure captions), and, in the reference list, the journals are not abbreviated, the binomial scientific name are not italicized, and [J] appears in each reference.

Response: We have modified the figure captions (page 25-29).

Comment 2: Also, it would be good if the superscript letters in the tables were explained in the table´s footnotes or figure caption.

Response: We have added explanations of the footnotes to the tables (line 336,458)

Round 3

Reviewer 1 Report

Dear authors,

thanks for your job!